# Fluid Shear Stress Modulates Inflammation in Breast Cancer Microenvironment

**DOI:** 10.3390/diseases13120402

**Published:** 2025-12-15

**Authors:** Abir Abdullah Alamro, Ohood Amin AlSuwaidi, Amani Ahmed Alghamdi, Saba Abdi, Atekah Hazzaa Alshammari, Reem Nasser Alotaibi

**Affiliations:** Department of Biochemistry, College of Science, King Saud University, P.O. Box 2455, Riyadh 11451, Saudi Arabia; aalamro@ksu.edu.sa (A.A.A.); ohood.alsuwaidi@riyadhpharma.com (O.A.A.); aalghamedi@ksu.edu.sa (A.A.A.); ahalshammari@ksu.edu.sa (A.H.A.); ralotaibi3@ksu.edu.sa (R.N.A.)

**Keywords:** fluid shear stress, breast cancer, anti-cancer effect, macrophage polarization, THP-1-like macrophages, MCF-7, tumor-associated macrophages

## Abstract

Background: Fluid shear stress (FSS) is a biomechanical force that can produce phenotypic changes in the cells that are directly in contact with the flow of fluid. Accumulating evidence indicates high FSS to possess the potential ability to prevent tumor development and suppress cancer growth. However, the exact mechanism of its antitumorigenic effects is still not clear. Objective: In this study, we aimed to investigate the effect of FSS on breast cancer microenvironment via macrophage modulation. Methods: We exposed THP-1 like-macrophages to different levels of FSS. The supernatant from THP1-like-macrophages after exposure to FSS was used as conditioned medium (FSS-CM). Subsequently, we analyzed human breast cancer cells, MCF-7, and endothelial cells, as well as HUVECs cultured with FSS-CM. Results: Study outcomes have demonstrated that low FSS-CM inhibited apoptosis as well as induced tumor migration in MCF-7 cells. Conversely, high FSS-CM promoted apoptosis, inhibited tumor migration, and induced G1-phase arrest in MCF-7 cells. Furthermore, low FSS-CM was found to promote proliferation of HUVECs. Conclusions: In conclusion, this study highlights the complex interplay between FSS and cancer cell behavior. Our findings provide in vitro evidence that high FSS exerts an anti-cancer effect by promoting THP-1-like macrophage polarization toward an anti-tumor phenotype, leading to increased apoptosis and reduced migration in MCF-7 cells. These results suggest that the modulation of macrophage polarization may underlie the therapeutic potential of high FSS in suppressing breast cancer progression.

## 1. Introduction

Female breast cancer has a high incidence rate and is the fifth leading cause of cancer related deaths globally [1]. The global burden of breast cancer is projected to increase significantly in the coming decades; by 2040, the number of new cases is expected to rise by over 40%, reaching approximately 3 million annually [2]. One of the foremost factors contributing to poor outcome of cancer is the metastatic spreading of tumor cells [3]. Traditionally, it was considered that genetic and epigenetic alterations within tumor cells essentially contribute to the metastatic dissemination of cancer cells from the primary tumor. However, currently it is well established to be a complex disease strongly driven by the intricate ecosystem surrounding tumor cells, which is termed as the tumor microenvironment (TME).

The components of TME, including several noncancerous stromal cells, the extracellular matrix (ECM), and biomechanical forces together, impact and control the metabolic alterations in cancer cells [4,5,6]. Specifically, tumor-associated macrophages (TAMs), one of the most abundant stromal cells in TME, have been implicated in prognosis of a variety of tumors [7,8]. During differentiation, TAMs exhibit phenotypic plasticity in response to microenvironmental cues. As a result, they can be functionally polarized into M1 (pro-inflammatory) or M2 (anti-inflammatory) phenotypes [9,10]. Growth factors and cytokines secreted by M2-like macrophages within the TME promote metastasis [11,12], whereas those released by M1-like macrophages exert tumor-suppressive effects [13]. Hence, investigating the involvement of each component of TME in regulating macrophage differentiation is crucial for cancer biology and therapy. While several studies have extensively focused on the role of chemical factors in macrophage polarization [11,14,15,16], relatively few have been designed to understand the role of biomechanical forces involved [17]. Research on the biomechanics of the TME is still in its infancy partially owing to past limitations in available technologies. Therefore, exact molecular mechanism for regulation of macrophage differentiation and function by biomechanical forces is not yet well elucidated. One of the noteworthy biomechanical cues that exert influence on cells within TME is the fluid shear stress (FSS) [18]. The FSS, defined as an internal frictional force between moving layers in laminar flow, can be understood as the force perpendicularly applied per unit area on a surface [19]. In addition to modulating macrophage differentiation within the TME, FSS influences the behavior of various stromal components [20]. Within tumors, cells experience a wide range of shear forces, from very low levels in interstitial space to higher magnitudes in vascular regions [21]. The persistently altered shear conditions in breast TME highlights the importance of investigating how FSS regulates macrophage function and how macrophage derived factors in turn affect cancer cell behavior [22].

In breast cancer cells, FSS has been shown to regulate stemness, survival, metastatic potential, and adhesion dynamics [23,24,25,26]. Although these studies were performed in simplified in vitro systems, they suggest that mechanical cues such as FSS may contribute to tumor cell adaptation within the broader TME. FSS has been revealed to cause cell death either by increasing the sensitivity of cells to cytokine induced apoptosis [27], enhancing oxidative stress [28], or by causing damage to cell membrane [29]. Thus, FSS plays a dual role in shaping the TME by influencing both TAM polarization and cancer cell growth and progression.

Despite its significance, the mechanistic basis of macrophage-cancer cell crosstalk under shear stress in the breast cancer microenvironment remains poorly understood, underscoring the need for further investigation to gain deeper insights into tumor biology. TAM-secreted factors are shaped by the biochemical and biophysical cues encountered during differentiation; thus, FSS-driven shifts in macrophage polarity may alter the release of mediators that critically regulate cancer cell growth and progression. The impact of FSS on breast cancer cells via macrophage modulation is still largely unknown, prompting the present investigation. THP-1, an acute monocytic leukemia cell line, is a widely used model of human macrophages. We investigated how FSS modulates THP-1 macrophage polarization and secretory profiles, and how these changes indirectly affect human breast cancer cells (MCF-7) and human umbilical vein endothelial cells (HUVECs) cell behavior. THP-1 cells were exposed to defined FSS levels for 24 h, after which the supernatants were collected as FSS-conditioned medium (FSS-CM) and used to treat MCF-7 and HUVECs to evaluate paracrine effects.

## 2. Materials and Methods

### 2.1. The Cell Culture and Differentiation

The HUVEC lines (ATCC Number CRL-1730™) and the human acute monocytic leukemia THP-1 cell lines (ATCC Number TIB-202™) were obtained from American Type Culture Collection (ATCC^®^), Manassas, VA, USA. Both cell lines were maintained in RPMI 1640 culture medium. MCF-7 cells were also obtained from ATCC^®^ (ATCC number HTB-22^TM^) and cultured in Dulbecco’s Modified Eagle Medium (DMEM). All cultures were supplemented with 10% fetal bovine serum along with 1% penicillin–streptomycin and incubated at 37 °C in a humidified 5% CO_2_ atmosphere.

The differentiation of THP-1 monocytes into macrophage-like phenotype was bought by using 100 nM of phorbol 12-myristate 13-acetate (PMA) for 3 days followed by 1 day in PMA-free medium and incubated at 37 °C with 5% CO_2_ concentration in air. Morphological changes in THP-1 monocytes following PMA treatment were analyzed by comparing undifferentiated cells versus differentiated cells under microscope.

### 2.2. Fluid Shear Stress Application

Following differentiation, THP-1 derived macrophages were seeded at an equal density of 1 × 10^6^ cells/mL to achieve approximately 80% confluence prior to FSS exposure. The cells were either exposed to orbital FSS or maintained under static conditions for 24 h. FSS exposure was performed using an orbital shaker (Thermo Fisher Scientific, Marietta, OH, USA) placed inside a humidified CO_2_ incubator (37 °C, 5% CO_2_). Static control macrophages were cultured in the same incubator under identical environmental conditions but without orbital motion, ensuring comparable cell numbers and matched parameters across all groups. The maximum shear stress (τ*_max_*) generated in the culture medium was calculated according to the equation [30]:$$\tau_{max}=a\sqrt{\rho \eta {\left(2\pi f\right)}^{3}}$$
where “*a*” is the orbital radius of rotation of the shaker (1.27 cm), “*ρ*” is the density of the culture medium (0.9973 g/mL), “*η*” is the viscosity of the medium (0.0101 poise), and “*f*” is the frequency of rotation (revolutions/second. Shaking speeds of 25, 45, 128, and 185 rpm produced estimated FSS levels of 0.5, 1.3, 6.1, and 10.6 dyne/cm^2^, respectively.

After 24 h, both FSS-exposed and static macrophages were processed in parallel. Cell viability and total cell number were quantified using a Vi-CELL™ XR 2.03 Cell Viability Analyzer (Beckman Coulter, Indianapolis, IN, USA) based on the trypan blue dye exclusion method, confirming no significant differences between groups. The results showed that >95% of cells remained viable at all FSS levels, comparable to the static (no shear) control. These findings confirm that the applied shear conditions were sub-cytotoxic and that the observed effects on MCF-7 and HUVECs are attributable to FSS-induced functional modulation of macrophages rather than differences in cell number or viability. Culture supernatants were then collected, centrifuged at 1500 rpm for 3 min to remove debris, and stored at −80 °C as FSS-conditioned medium (FSS-CM) or static-conditioned medium (static-CM), to be used subsequently for stimulation of HUVEC and MCF-7 cells.

### 2.3. THP-1-like Macrophages Profiling Assay

Based on pilot dose–response testing, no significant differences in macrophage viability or canonical activation markers were observed between 10.6 dyne/cm^2^ and 11.6 dyne/cm^2^. Therefore, an FSS of 11.6 dyne/cm^2^ was selected for macrophage phenotyping to capture a maximal activation profile, whereas 10.6 dyne/cm^2^ was used as the highest shear stress in all other experiments to ensure reproducibility and maintain cell integrity during prolonged exposure. THP-1-derived macrophages were exposed to 11.6 dyne/cm^2^ for 12 and 24 h. Following shear application, the cells were centrifuged at 1500 rpm for 5 min, and the supernatant was removed. To prevent non-specific binding, fragment crystallizable (F-c) receptors were blocked by resuspending the cell pellet in 100 µL of 10% Fc receptor blocking reagent (Miltenyi Biotec, Auburn, CA, USA) prepared in 1% bovine serum albumin (Sigma-Aldrich, Darmstadt, Germany), followed by incubation for 20 min at room temperature.

For detection of cell surface markers (CD49c, CD38, CD47) cells were stained with the corresponding antibodies and incubated for 1 h at 4 °C in dark. To detect the intracellular markers β-tubulin cells were permeabilized with 20 µL of 0.1% tween (BIO-RAD, Watford, UK) and further incubated for 20 min at room temperature. Following incubation, cells were stained with the intracellular antibodies and incubated for 1 h at 4 °C in dark. Then, cells were centrifuged at 1500 rpm for 3 min, supernatant was discarded, and cells were resuspended with 300 µL Phosphate-Buffer Saline (PBS). BD FACS Canto-II flow cytometry machine (BD Biosciences, San Jose, CA, USA) was used for running samples and BD FACSDiva^TM^ Software Version 6 for data analysis. All antibodies (Table 1) were purchased from BD Biosciences, USA.

### 2.4. Conditioned Media Treatment

MCF-7 and HUVEC lines were seeded to reach approximately 80% confluence within 24 h. The cells were then cultured for 48 h in complete medium supplemented with 50% CM obtained from THP-1-derived macrophages exposed to FSS at 0.5, 1.3, 6.1, or 10.6 dyne/cm^2^ (FSS-CM) or from macrophages maintained under static conditions (static-CM) in a humidified incubator (37 °C, 5% CO_2_). Equal numbers of THP-1-derived macrophages were seeded and maintained across all conditions to ensure comparable cell density at CM collection.

### 2.5. Cell Apoptosis Assay

Apoptosis in MCF-7 cells was assessed using the Hoechst 33342/Propidium Iodide (PI) double-staining apoptosis detection kit (Thermo Fisher Scientific, USA). MCF-7 cells were cultured for 48 h in complete medium supplemented with 50% CM obtained from THP-1-derived macrophages exposed to different levels of FSS: 0.5, 1.3, 6.1, or 10.6 dyne/cm^2^) or from static macrophages (static-CM, control). After the 48 h treatment period, both adherent and floating cells were collected to ensure inclusion of any detached apoptotic cells. The combined cell suspension was washed twice with phosphate-buffered saline (PBS). Cells were then resuspended in 100 µL of Hoechst 33342 solution (10 µg/mL in PBS) and incubated at 37 °C for 10 min. After centrifugation at 1200 rpm for 3 min, the supernatant was discarded, and the cells were resuspended in PI solution (20 µg/mL in PBS) and mixed thoroughly. Following 15 min of incubation at room temperature, the cells were immediately analyzed by flow cytometry.

### 2.6. Cell Cycle Analysis

MCF-7 cells treated with different FSS-CM (FSS of 0.5, 1.3, 6.1, and 10.6 dyne/cm^2^) were harvested and collected in FACS tubes and washed with PBS. Then, the supernatant was discarded, and cells were fixed with 500 μL of 70% cold ethanol (SIGMA-ALDRICH, Germany). To ensure fixation of all cells and minimize clumping, ethanol was added dropwise to the cell pellet while vortexing. Cells were fixed overnight at 4 °C. After permeabilization, 2–3 mL of PBS was added per tube, and cells were centrifuged at 3000 rpm for 5 min. Supernatant was discarded and cells were resuspended in 1mg/mL RNase (Thermo Fisher Scientific, USA) diluted in PBS. Then, cells were stained with 2 μg/mL PI. After 30 min of incubation at room temperature, cells were immediately analyzed by flow cytometry. The percentages of cells with G1, S, and G2 DNA content were used to determine the cell cycle rate of the cell population.

### 2.7. Cell Imaging

MCF-7 cells were seeded to reach 60% of confluence in 24 h. Next, the old media was discarded, and cells were cultured with 50% of different FSS-CM (FSS of 0.5, 1.3, 6.1, and 10.6 dyne/cm^2^) or static-CM and incubated for 48 h in humidified incubator (37 °C—5% CO_2_). Then, media was removed and changed by 100 μL of 10 μg/mL Hoechst 33342 staining solution diluted in PBS and incubated in the dark for 15 min at 37 °C. The Hoechst solution was discarded, and cells were rinsed with 1× PBS. Fluorescent microscopy images were obtained using EVOS cell imaging system (Thermo Fisher Scientific, USA) with a 40-fold magnification.

### 2.8. Migration Assay

Migration of MCF-7 was measured using wound healing scratch assay. The cells were grown to reach 100% confluence. Old media was discarded, and cells were wounded by scratching with a sterile 200 μL pipette tip. Excess debris and cell clumps were cleared away by rinsing with PBS. Then, cells were cultured with 50% of different FSS-CM (FSS of 0.5, 1.3, 6.1, and 10.6 dyne/cm^2^) or static-CM. To observe wound closure, plates were monitored for 24 h using a time-lapse imaging microscopy (EVOS^®^ FL Auto Imaging System, Thermo Fisher scientific, USA). Time-lapse imaging microscopy was equipped with ‘Onstage Incubator’ where temperature and the atmosphere were kept at 37 °C and 5% CO_2_, respectively. Phase-contrast pictures of the wound margins were taken using a 10× objective magnification every 4 h throughout the 24 h experiments. The recorded images were analyzed using ImageJ software (version 1.54h) (W.S. Rasband, National Institutes of Health, Bethesda, MD, USA). Initially, the wound area was measured. Subsequently, percent closure and area under the curve (AUC) were calculated.

### 2.9. Cell Metabolic Assay

HUVECs were seeded in 96-well plates to reach 80% confluence within 24 h. The next day, the medium was replaced with 50% conditioned medium derived from THP-1-like macrophages exposed to different levels of FSS-CM (0.5, 1.3, 6.1, and 10.6 dyne/cm^2^) or from static macrophages (static-CM, control), and cells were incubated for 48 h at 37 °C in a humidified incubator with 5% CO_2_. After treatment, the medium was removed and replaced with 50 μL phenol red-free medium containing 2.5 μL MTT (3-(4,5-dimethylthiazol-2-yl) 2,5-diphenyl tetrazolium bromide) reagent (5 mg/mL in PBS; Invitrogen, Carlsbad, CA, USA). Cells were incubated in the dark for 3 h at 37 °C. Subsequently, the medium was discarded, and the resulting formazan crystals were solubilized in 100 μL DMSO. The plate was shaken for 3 min on an orbital shaker, and absorbance was measured at 560 nm using a SpectraMax^®^ M5 Microplate Reader (Molecular Devices, San Jose, CA, USA).

### 2.10. Statistical Analysis

All data were analyzed using GraphPad Prism 8.0 (www.graphpad.com, San Diego, CA, USA). The data are presented as means ± SEM, derived from a minimum of three independent replicated experiments. Statistical comparison was performed using a two-tailed *t* test. *p* < 0.05 was considered significantly different.

## 3. Results

### 3.1. This THP-1 Differentiation

The undifferentiated THP-1 monocytes (Figure 1A) after PMA treatment became adherent and acquired a characteristic spread morphology (Figure 1B). This suggested that THP-1 monocytes cells acquired a macrophage-like phenotype after PMA stimulation.

### 3.2. THP-1-like Macrophages Profiling (Cell Marker Expression Analysis)

FSS (11 dyne/cm^2^, 12 h and 24 h exposure) induced cellular phenotypic changes in THP-1-like macrophages were determined by examining the abundance of CD49c, CD38, CD47, and β-Tubulin, compared to the control group (0 dyne/cm^2^) (Figure 2). As compared to control cells there was a significant increase (*p* < 0.01) in expression of CD49c, CD38, and β-Tubulin, whereas a significant decrease (*p* < 0.05) in expression of CD 47 in THP-1-like macrophage following 12 h shear stress application was observed. When the shear stress application was performed after 24 h, there was further increase in expression of CD49c, as well as of CD47 compared to expression after 12 h application. The expressions of CD 38 and β-Tubulin, however, showed reduction after 24 h shear stress application, compared to the rise observed after 12 h shear stress application.

### 3.3. FSS-Induced MCF-7cell Death

MCF-7 cells were cultured with 50% FSS-CM for 48 h. Flow cytometric analysis using Hoechst/PI double staining was performed to evaluate viable, apoptotic, and non-viable (late apoptotic/necrotic) cell populations. As shown in Figure 3A, no significant differences in the proportion of dead cells were observed across FSS conditions, although a trend toward reduced number of dead cells was noted at higher shear stress levels. Similarly, no significant differences in the proportion of necrotic cells were observed across FSS conditions, although a trend toward reduced number of necrotic cells was noted at higher shear stress levels (Figure 3B). The percentage of apoptotic cells (Figure 3C) was significantly lower than the control at 0.5 dyne/cm^2^ and showed a progressive, though statistically non-significant, increase with rising FSS intensity.

Changes were also observed in the nuclear morphology of FSS-CM exposed cells (Figure 4). The nuclei of the control MCF-7 cells were homogeneously stained and regularly shaped whereas at 1.3 dyne/cm^2^, early apoptotic nuclei showed condensed chromatin. While at 6.1 and 10.6 dyne/cm^2^, the nuclei of MCF-7 cells were cleaved and apoptotic bodies formed were clearly visible.

### 3.4. FSS-CM Induces G1-Phase Cell Cycle Arrest in MCF-7

Results (Figure 5) demonstrated that the treatment of MCF-7 cells with CM derived from macrophages exposed to higher FSS (6.1 dyne/cm^2^) led to a statistically significant (*p* = 0.001) accumulation of MCF-7 cells in the G1 phase, when compared to cells exposed to static-CM suggesting cell-cycle arrest. MCF-7 cells exposed to low FSS-conditioned media (0.5 dyne/cm^2^) show a significantly lower fraction of cells in G1 phase compared to cells exposed to high FSS-conditioned media (6.1 and 10.6 dyne/cm^2^). Further, MCF-7 cells cultured in low FSS-conditioned medium (0.5 dyne/cm^2^) exhibited a significantly lower proportion of cells in G1 compared with cells exposed to FSS-CM 6.1 (*p* = 0.006) and FSS-CM 10.6 (*p* = MCF-7 cells cultured in low FSS-conditioned medium (0.5 dyne/cm^2^) exhibited a significantly lower proportion of cells in G1 compared with cells exposed to FSS-CM 6.1 (*p* = 0.006) and FSS-CM 10.6 (*p* = 0.05).

### 3.5. Migration Assay

A wound healing (scratch) assay was performed to evaluate MCF-7 cell migration. Cells were treated with 50% FSS-CM derived from THP-1-like macrophages exposed to shear stresses of 0.5, 1.3, 6.1, or 10.6 dyne/cm^2^. MCF-7 cells treated with 50% conditioned medium from static THP-1-like macrophages (static-CM) served as the control. Images were captured over a period of 24 h to observe changes in cellular migration rates under the effect of varying levels of FSS. Raw scratch assay images (Figure 6) were quantified using ImageJ software and the analyzed results are presented in Figure 7.

Wound healing assays revealed that MCF-7 cell migration was differentially regulated by macrophage-CM generated under distinct FSS levels. Low FSS-CM (0.5 and 1.3 dyne/cm^2^) promoted complete wound closure over 24 h and statistically enhanced the summed AUC as compared to the static-CM controls. In contrast, highest FSS-CM (10.6 dyne/cm^2^) not only inhibited migration compared with low FSS-CM but also statistically reduced the summed AUC below that of static-CM controls.

### 3.6. HUVEC Metabolic Activity in Response to FSS-Conditioned Media

Figure 8 shows that HUVEC metabolic activity was significantly increased following treatment with low FSS-CM (0.5 and 1.3 dyne/cm^2^) and decreased in response to high FSS-CM (10.6 dyne/cm^2^) compared with static-CM controls. These results indicate that FSS-CM alters HUVEC metabolic activity, which may correlate with changes in proliferation or cell viability, although it does not provide a direct measure of cell proliferation.

## 4. Discussion

The present study demonstrates that FSS can modulate macrophage function in a manner that profoundly influences the behavior of the neighboring tumor and endothelial cells. CM obtained from THP-1-derived macrophages exposed to increasing levels of FSS (0.5–10.6 dyne/cm^2^) produced distinct and opposing cellular responses. While CM generated under low FSS conditions inhibited apoptosis and promoted migration of MCF-7 breast cancer cells, high FSS-CM enhanced apoptosis, suppressed tumor migration, and induced G1-phase cell cycle arrest. In addition, low FSS-CM stimulated HUVEC proliferation, indicating shear-dependent modulation of angiogenic potential. These findings reveal a previously underexplored mechanobiological pathway through which the magnitude of shear stress experienced by macrophages differentially regulates their secretory profile, thereby influencing both tumor progression and vascular responses.

The TME of mammary gland is characterized by a complex interplay of dynamic mechanical factors, including FSS, compression, tension, and variations in ECM stiffness [31,32,33]. Among these, FSS is a particularly potent biophysical signal that drives phenotypic and functional changes in cells exposed to fluid flow [34,35]. Yet, the mechanisms through which these mechanical forces regulate macrophage–tumor interactions remain incompletely understood. The present findings extend this understanding by demonstrating that macrophage-secreted factors, conditioned by distinct FSS magnitudes, can differentially affect cancer cell survival and migration. This work thus provides new insight into the mechanobiological regulation of macrophage–tumor interactions and lays a foundation for exploring how interstitial flow and shear-mediated signaling could be targeted to modulate tumor progression.

Fluidic stimuli within the human body can be broadly categorized into two types as follows: those arising from flow in blood and lymphatic vessels (vascular and lymphatic flow, respectively) and those generated by fluid movement through tissues, termed interstitial flow [36]. The magnitude of FSS experienced by cells within the TME varies considerably, ranging from more than 10 dyne/cm^2^ near perfused vasculature [37,38] to less than 0.1 dyne/cm^2^ in regions of interstitial flow [39,40,41,42]. This mechanical heterogeneity plays a crucial role in shaping cellular responses, influencing not only cell morphogenesis but also the pathogenesis of various diseases, including cancer [43]. The mechanical stiffness of healthy and cancer cells differs significantly, which can influence the transduction of forces regulating the cell’s invasive behavior. Previous investigations have largely focused on the direct impact of FSS on tumor cells. For instance, Pan and colleagues demonstrated that low FSS promotes migration, invasion, and drug resistance in triple-negative breast cancer cells, whereas high FSS exerts the opposite effects [44]. Similarly, Yu et al. reported that exposure of hepatocellular carcinoma (HepG2) cells to low FSS (1.4 dyne/cm^2^) enhanced their migratory capacity through activation of the integrin-mediated signaling pathway [45]. In contrast, our study highlights the indirect role of FSS, showing that shear stress-conditioned macrophages can differentially influence tumor and endothelial cell responses, thus revealing an additional mechanobiological layer in cancer regulation.

In solid tumors, such as those of breast, TAMs are considered as the most relevant stromal cells in relation to carcinogenesis, often being regarded as potential biomarkers for breast cancers [35,36]. These plastic cells can polarize toward a pro-inflammatory M1 phenotype, or an immunosuppressive M2 phenotype based on the stimuli they encounter in their surroundings [11]. Consistent with this interpretation, exposure of THP-1-derived macrophages to FSS induced distinct alterations in surface marker expression (CD49c, CD38, CD47) and cytoskeletal organization (β-tubulin), indicating an activated, motile, and secretion-competent phenotype. These changes suggest that shear stress promotes macrophage activation and functional polarization toward a pro-inflammatory-like state, consistent with previous reports of shear-induced immune activation. Adhesion molecules on leukocytes mediate interactions with other cells or the ECM and participate in mechanotransduction, converting mechanical cues into biological responses [33]. CD49c (α3 integrin) facilitates ECM binding and is upregulated during monocyte-to-macrophage differentiation, serving as a macrophage marker [46]. CD38 and CD47 reflect microenvironmental regulation of macrophage phenotype, with CD38 being a hallmark of pro-inflammatory (M1-like) activation [47,48]. The observed upregulation of CD49c, CD38, and CD47 in THP-1-derived macrophages after 24 h exposure to high FSS (11.6 dyne/cm^2^) indicates shear-induced phenotypic reprogramming toward an M1-like pro-inflammatory state. Previously, shear stress has been shown to induce a pro-inflammatory transcriptional program in human monocyte-derived macrophages, characterized by increased cytokine production and enhanced motility [49]. Thus, our findings reinforce the concept that shear stress can act as a potent biophysical cue driving macrophage differentiation, providing a mechanistic basis for the observed downstream effects of FSS-conditioned medium on tumor and endothelial cell behavior.

The functional relevance of this macrophage reprogramming was further reflected in the distinct biological effects of their CM on recipient MCF-7 cells. CM from macrophages under low FSS (0.5 dyne/cm^2^) appears to enhance MCF-7 cell survival. Although apoptotic percentages rose with increasing FSS, the trend did not reach significance and therefore suggests, rather than proves, a dose-dependent reversal of this effect. These data support the idea that low mechanical stress can program macrophages to produce survival-promoting signals that favor tumor persistence. Consistent with our observations, Li et al. reported that interstitial fluid flow, corresponding to low shear stress, induces macrophage polarization toward an M2 phenotype [50]. This supports our hypothesis that low mechanical stress conditions can program macrophages to release pro-survival mediators that favor tumor persistence.

Apoptosis of MCF-7 cells following exposure to FSS-CM was further validated by fluorescence microscopy, which revealed characteristic nuclear condensation and fragmentation. The number of viable cells decreased progressively with increasing FSS intensity, indicating an inverse relationship between shear stress magnitude and cell survival. These observations suggest that macrophage-derived cytotoxic mediators present in high-magnitude FSS-CM (6.1 dyne/cm^2^ and 10.6 dyne/cm^2^) contribute to enhanced apoptosis of MCF-7 cells. This indicates that elevated FSS may indirectly exert anti-proliferative effects through modulation of macrophage function and alteration of their secretory profile.

Genes that are involved in cell cycle progression regulate apoptosis [51]. Subsequently, flow cytometry analysis of PI-stained nuclei was performed to further explore whether the inhibitory effect of high FSS on MCF-7 cells is due to its induction of cell-cycle arrest. Our data suggest that exposure of macrophages to elevated shear stress (6.1 dyne/cm^2^) reprograms their secretory activity in a manner that promotes G_1_-phase arrest in MCF-7 cells. This observation is supported by findings of Son et al., who showed that shear stress programs THP-1 derived macrophages towards pro-inflammatory (M1-like) phenotype [52]. Earlier, a study by Engström et al. showed that CM from M1, but not M2, macrophages inhibited tumor cell proliferation and induced G_0_/G_1_ arrest [53]. These parallels indicate that shear stress may drive macrophages toward an M1-like, anti-proliferative phenotype capable of modulating tumor cell cycle progression. Taken together, these findings support the emerging concept that biomechanical cues within the TME can reprogram macrophages toward an anti-proliferative phenotype, thereby influencing tumor progression. Specifically, FSS appears to modulate macrophage–cancer cell crosstalk in a manner that favors growth inhibition and promotes apoptotic signaling in tumor cells.

Our scratch wound-healing results reveal a biphasic effect of macrophage secretions under varying FSS: low FSS-CM (0.5–1.3 dyne/cm^2^) enhanced MCF-7 migration, whereas high FSS-CM (6.1–10.6 dyne/cm^2^) inhibited it. This indicates that macrophage-secreted factors vary markedly with the magnitude of mechanical stimulation. Low shear may favor a pro-migratory, pro-tumor phenotype, consistent with reports that low FSS enhances stem-like and metastatic traits in breast cancer cells [54]. Conversely, higher FSS suppressed tumor cell motility, possibly by stimulating macrophage secretion of cytotoxic or anti-migratory mediators such as TNF-α, IL-1β, or reactive species. Although cytokine levels were not directly assessed in this study, these context-dependent, biphasic effects align with previous reports describing biomechanical regulation of cancer progression [52,54]. Collectively, our results support a model in which the magnitude of FSS experienced by TAMs determines their secretome and, consequently, the pro- or anti-migratory behavior of adjacent breast cancer cells. Mapping the responsible cytokines/enzymes will clarify whether the effect is driven mainly by chemotactic signals or by cytotoxic mediators.

Notably, high FSS when applied directly to tumor cells has been reported to induce tumor cell apoptosis rather than necrosis [20,24]. Higher interstitial flow rate is also an independent predictor of poor prognosis in cancer patients [55]. Interestingly, not all cancer cell types respond uniformly to shear stress [56,57,58]. Thus, our study adds to a growing body of evidence that the impact of FSS on tumor progression is context-dependent, determined by cell type, shear magnitude, and microenvironmental interactions. Overall, these findings suggest that modulation of hemodynamic forces in the TME could represent a previously underappreciated strategy to potentiate anti-tumor immune responses.

Endothelial metabolic activity is a central determinant of angiogenesis, which sustains tumor growth and metastasis [59,60]. In this study, HUVEC metabolic responses to FSS-CM generated under varying shear stresses revealed that macrophage polarization under FSS critically modulates endothelial function. High FSS-CM (10.6 dyne/cm^2^) significantly reduced HUVEC metabolic activity, implying that shear-exposed macrophages adopt an anti-tumor, anti-angiogenic phenotype, whereas low FSS-CM (0.5–1.3 dyne/cm^2^) enhanced activity, consistent with a pro-angiogenic state. These results support previous observations that mechanical forces regulate macrophage polarization and, in turn, endothelial behavior [61]. The suppressive effect of high FSS-CM is likely mediated by a pro-inflammatory macrophage secretome enriched in cytokines such as TNF-α and IL-1β, which have been shown to inhibit endothelial proliferation and angiogenic signaling [62,63]. Collectively, these findings highlight the role of shear stress in regulating TAM–endothelial crosstalk and its potential impact on tumor angiogenesis.

Depending upon the type of FSS the effect on cells may change [64]. Our current research was limited by the fact that we analyzed the effects of only orbital FSS, thus it would be interesting to design similar experiments in future where effects of laminar FSS on breast cancer microenvironment may be studied as well. Although the direct effects of FSS on MCF-7 cells were limited in our system, the potential for shear-induced macrophage activation to suppress tumor cell viability highlights a novel mechanobiological interaction. We speculate that high shear-exposed macrophages undergo phenotypic reprogramming, potentially toward an M1-like state, resulting in the secretion of cytotoxic cytokines, reactive oxygen species, or death ligands (e.g., FasL, TRAIL) that trigger apoptosis in neighboring MCF-7 cells, and thereby exhibits anti-proliferative effect. Future studies will be essential to identify the molecular signatures of macrophages exposed to shear stress and to pinpoint the specific mediators driving the observed cytotoxicity, such as reactive oxygen species or death ligands like FasL and TRAIL. Understanding these mechanisms could pave the way for harnessing mechanical forces as modulators of immune–tumor interactions. Additionally, integrating these foundational findings into in vivo models will be critical to assess how fluid flow and flow-altering therapies influence tumor invasion. Such research has the potential to open new frontiers in cancer biology and inspire innovative therapeutic approaches.

Overall, our findings support the notion that low shear stress fosters a pro-tumorigenic and pro-angiogenic microenvironment, whereas high shear stress tends to suppress these processes. The use of FSS-CM derived from shear-exposed THP-1-like macrophages revealed that macrophage-secreted factors can differentially influence tumor and endothelial cell behavior in a force-dependent manner. These results underscore the mechanosensitive nature of macrophages and their pivotal role in shaping breast tumor progression and angiogenesis.

From a translational perspective, the magnitude-dependent modulation of macrophage secretions that regulate MCF-7 apoptosis and HUVEC activity provides new insight into how mechanical cues govern TME dynamics. Targeting macrophage mechanotransduction pathways or their soluble mediators may offer novel therapeutic opportunities to control tumor growth and angiogenesis. Furthermore, incorporating shear stress parameters into preclinical breast tumor models could enhance their physiological relevance for drug testing and biomarker discovery, thereby advancing the development of more effective, microenvironment-informed cancer therapies.

## 5. Conclusions

In conclusion, we demonstrated that high FSS induces apoptosis and G1-phase arrest in MCF-7 breast cancer cells and inhibits the proliferation of endothelial cells (HUVECs), collectively contributing to an anti-tumor effect. Importantly, our findings reveal a novel mechanism whereby high FSS activates macrophages, which in turn modulate cancer cell survival and endothelial function, likely through changes in their secretory profile. These results highlight the critical role of biomechanical forces in regulating immune–tumor interactions and suggest that modulating FSS may offer a promising therapeutic approach. Further studies are needed to clarify the specific molecular mediators and pathways involved in TAM polarization under shear stress to support the development of FSS-based strategies for breast cancer treatment.

## Figures and Tables

**Figure 1 diseases-13-00402-f001:**
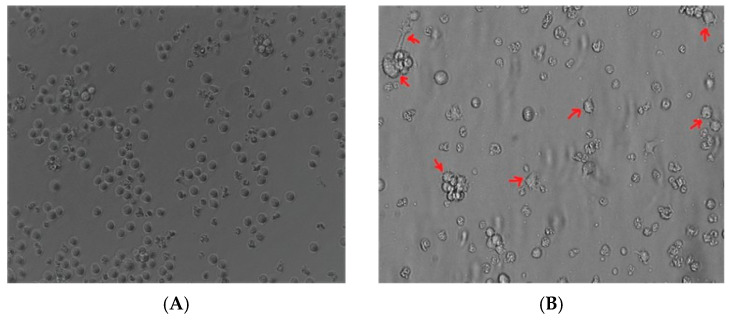
Induction of the macrophage-like phenotype in THP-1 cells. THP-1 cells were differentiated by PMA treatment. Morphological changes were analyzed by comparing (**A**) undifferentiated cells versus (**B**) differentiated cells. Arrows indicate typical “spreading.”

**Figure 2 diseases-13-00402-f002:**
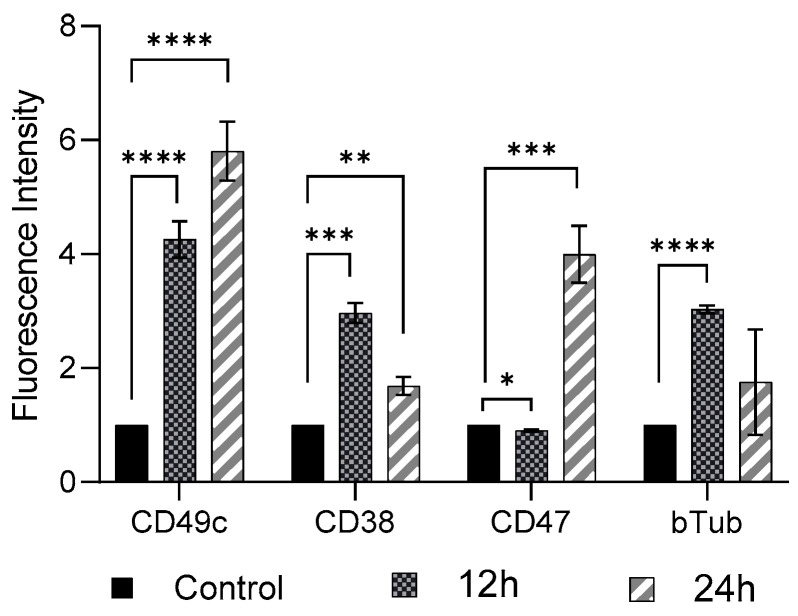
Cell marker expression on THP-1-like macrophages after FSS application of 11.6 dyne/cm^2^ for 12 h and 24 h. Cells were subjected to flow cytometry analysis at the indicated time-points. All results were confirmed in triplicate in three independent experiments under the same experimental conditions. Statistical differences in comparison to the control are indicated by asterisks (* *p* ≤ 0.05, ** *p* ≤ 0.01, *** *p* ≤ 0.001, **** *p* ≤ 0.0001).

**Figure 3 diseases-13-00402-f003:**
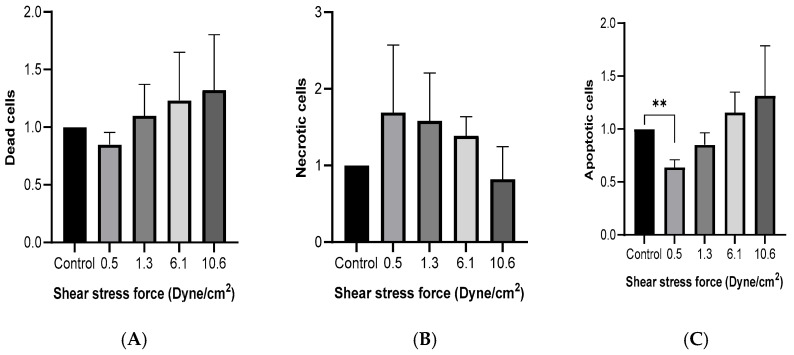
Fluid shear stress-induced cell death in MCF-7 cells. MCF-7 cells were cultured with 50% of FSS-CM of different magnitudes (0.5, 1.3, 6.1, and 10.6 dyne/cm^2^) (**A**) Cytotoxicity effect of FSS. (**B**) Effect of FSS on apoptosis. (**C**) Effect of FSS on necrosis. Dead, apoptotic, and necrotic cells were quantified by flow cytometry after staining with Hoechst and PI (Mean ± SEM, n = 3). ** *p* ≤ 0.01 as compared to the control. Control: MCF-7 cells cultured with 50% of static-CM.

**Figure 4 diseases-13-00402-f004:**
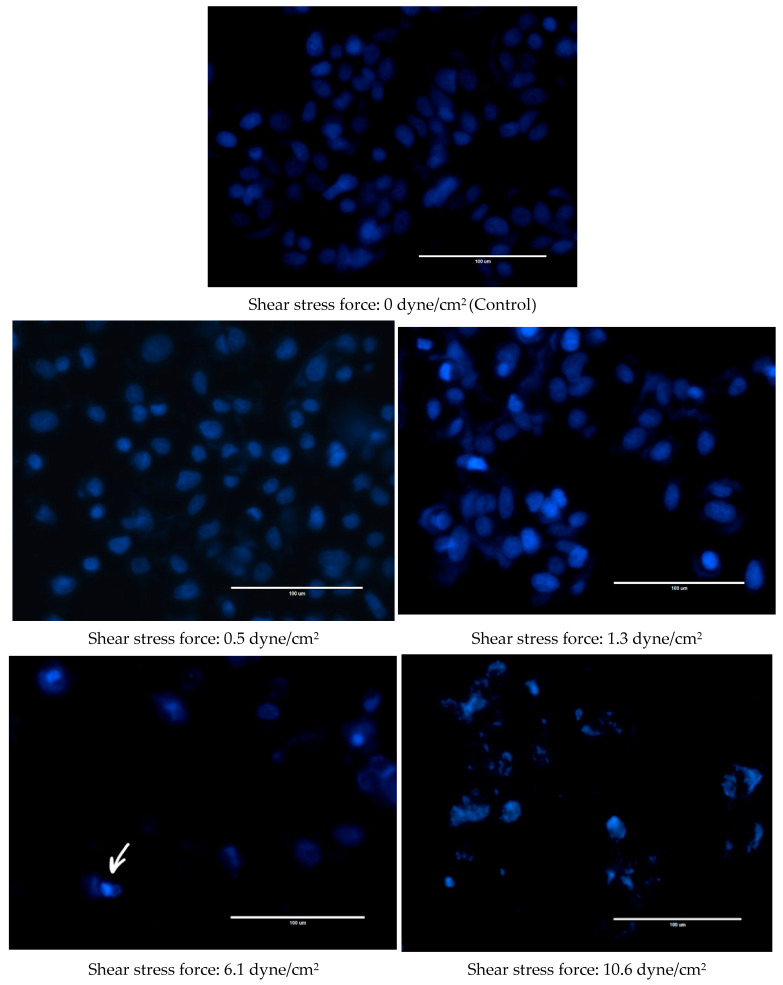
Fluid shear stress-induced apoptosis in MCF-7 cells. Representative Hoechst 33342 staining images of MCF-7 cells cultured with 50% of FSS-CM of different magnitudes (0.5, 1.3, 6.1, and 10.6 dyne/cm^2^) at magnification 40× (scale bar: 100 μm). Apoptotic bodies were observed both at 6.1, and 10.6 dyne/cm^2^ shear stress force. The arrow highlights apoptotic bodies formed in 6.1 dyne/cm^2^ panel as a representative example. Control: MCF-7 cells cultured with 50% static-CM. (shear stress of 0.0 dyne/cm^2^).

**Figure 5 diseases-13-00402-f005:**
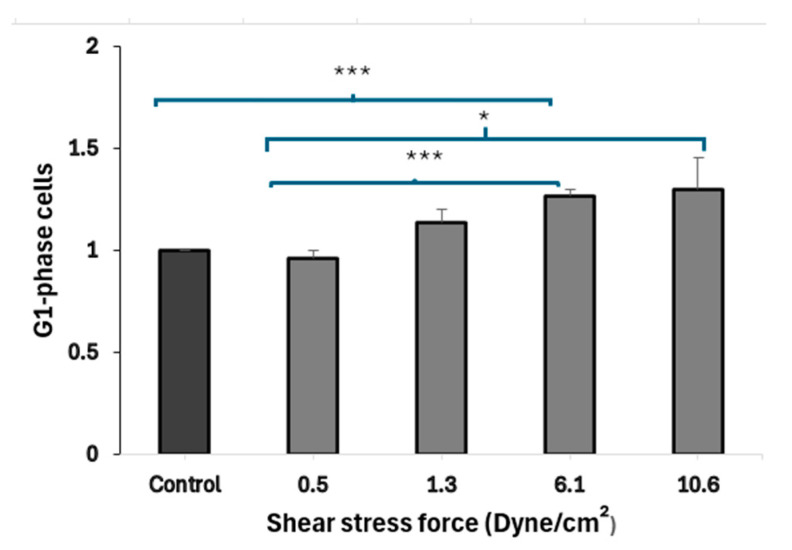
Fluid shear stress–induced G_1_-phase arrest in MCF-7 cells. Conditioned media (CM) from macrophages exposed to increasing FSS levels (0.5–10.6 dyne/cm^2^) were used to treat MCF-7 cells. CM from high FSS (6.1 dyne/cm^2^) significantly increased the G_1_-phase population as compared to control, indicating cell-cycle arrest. Data represent mean ± SEM (n = 3). * *p* ≤ 0.05, *** *p* ≤ 0.001 vs. control (MCF-7 cells treated with 50% static-CM).

**Figure 6 diseases-13-00402-f006:**
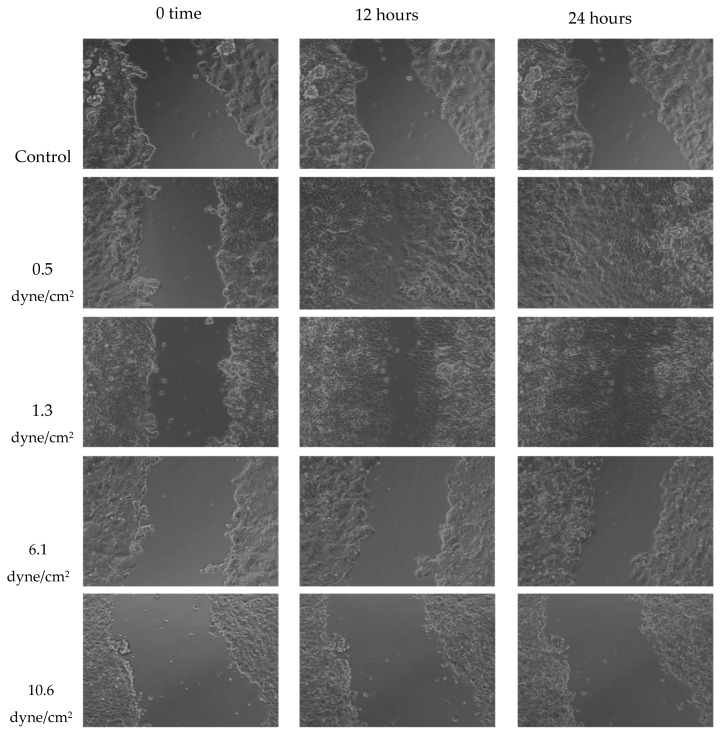
Representative scratch assay images for MCF-7 cells exposed to 50% of FSS-CM at different magnitudes (0.5, 1.3, 6.1, and 10.6 dyne/cm^2^) over 24 h period. The lowest FSS force images show a completely “healed” scratch after 24 h. Control: MCF-7 cells were cultured with 50% of static-CM over 24 h period.

**Figure 7 diseases-13-00402-f007:**
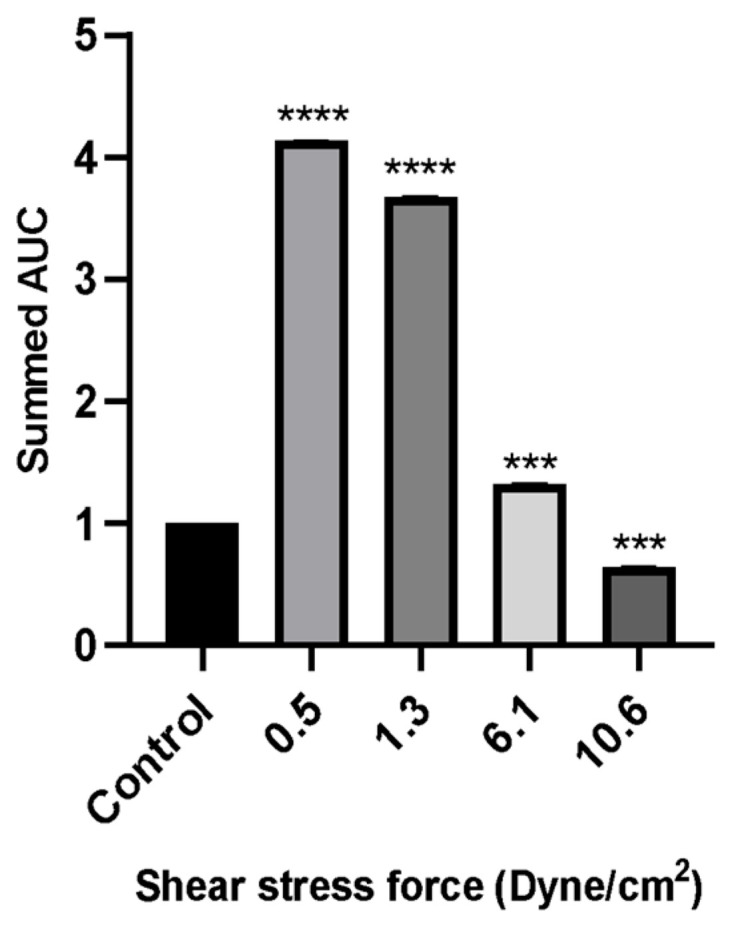
The wound healing rate of MCF-7 cells evaluated by scratch assay over a period of 24 h incubation with 50% of FSS-CM at different magnitudes (0.5, 1.3, 6.1, and 10.6 dyne/cm^2^). Control: MCF-7 cells were cultured with 50% static-CM for 24 h. High FSS-CM (10.6 dyne/cm^2^) significantly reduced MCF-7 migration compared with static-CM (control), as reflected by a lower summed AUC, indicating active inhibition of cell motility by shear stress-polarized macrophages, whereas low FSS-CM (0.5–1.3 dyne/cm^2^) promoted migration. Statistical differences in comparison to the control are indicated by asterisks (*** *p* ≤ 0.001, **** *p* ≤ 0.0001).

**Figure 8 diseases-13-00402-f008:**
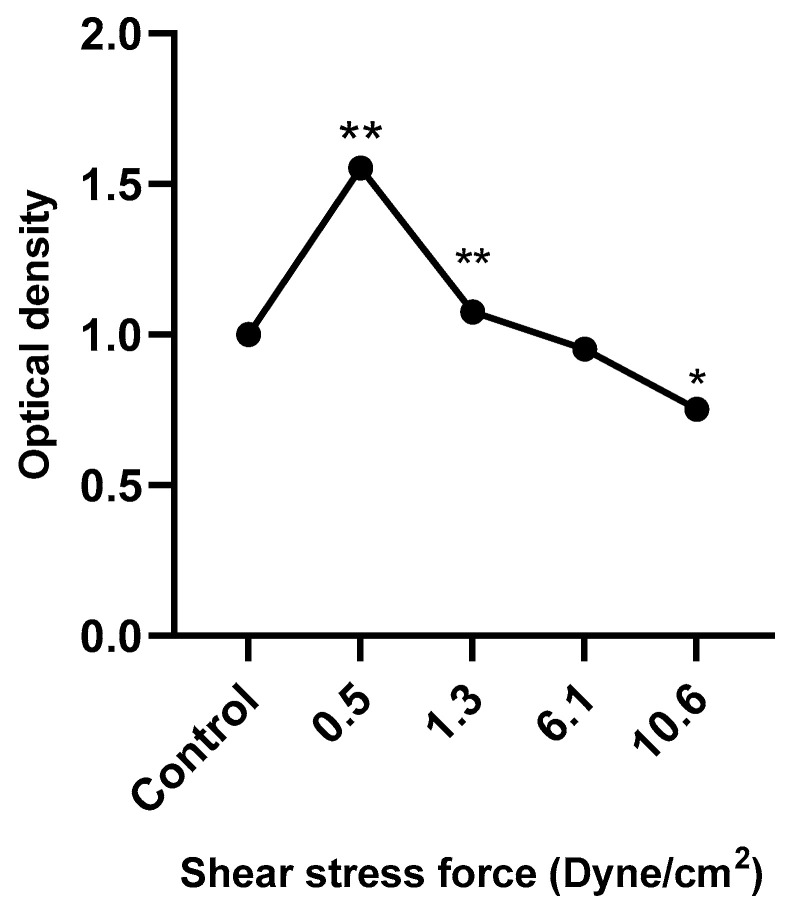
HUVECs were cultured with 50% of FSS-CM at different magnitudes (0.5, 1.3, 6.1, and 10.6 dyne/cm^2^) and incubated for 48 h. Control: HUVECs cultured with 50% of static-CM and incubated for 48 h. Low FSS-CM increased, whereas high FSS-CM decreased HUVECs metabolic activity. The values of cell proliferation are calculated in relation to control (* *p* ≤ 0.05, ** *p* ≤ 0.01).

**Table 1 diseases-13-00402-t001:** Anti-human conjugated antibodies for macrophage characterization.

Marker	Catalog Number	Conjugated Dye
CD 49c	556025	PE
CD 38	556025	PERCP-Cy 5.5
CD 47	556046	PE
β Tubulin	560394	PERCP-Cy 5.5

## Data Availability

The data included in this article are available upon reasonable request.

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
