# Peer review of "Fluid Shear Stress Modulates Inflammation in Breast Cancer Microenvironment"

_diseases, 2025, doi:10.3390/diseases13120402_

Round 1
Reviewer 1 Report (Previous Reviewer 2)
Comments and Suggestions for Authors
The experimental work submitted for review is devoted to studying the effect of FSS on the breast cancer microenvironment through macrophage modulation. The authors found that high levels of FSS have an anti-cancer effect by releasing components similar to THP-1 macrophages, which potentially mimic the polarization of tumor-associated macrophages in accordance with the antitumor phenotype. There are the following questions and suggestions about this work: 1. The "conclusion" section should be highlighted in the abstract. 2. The section "Discussion" should begin (first paragraph) with the results obtained in this paper, summarizing their essence and indicating the novelty and priority. 3. In the "discussion" section (second paragraph), it is unclear why the phrase "Figure 40. The cells exposed to vascular blood flow can encounter much higher fluid 378 shear stress [41]." 4. What practical significance do the results have? How realistic is it to implement them in practical oncology? This should be indicated at the end of the "discussion" section. 5. Is there genetic evidence linking polymorphisms involved in FSS with breast cancer? This point should be reflected in the discussion of the work.
Author Response
Comment 1: The "conclusion" section should be highlighted in the abstract.
Response 1: We thank the reviewer for this suggestion. The “Abstract” has been revised to include a clearly defined conclusion section summarizing the key findings and their implications. The revised statement for conclusion of abstract now appears in the last seven lines of the ‘Abstract” (Lines 24-30) and reads as follows:
Conclusion: In conclusion, this study highlights the complex interplay between FSS and cancer cell behavior. Our findings provide in vitro evidence that high FSS exerts an anti-cancer effect by promoting THP-1-like macrophage polarization toward an anti-tumor phenotype, leading to increased apoptosis and reduced migration of MCF-7 cells. These results suggest that modulation of macrophage polarization may underlie the therapeutic potential of high FSS in suppressing breast cancer progression.
Comment 2: The section "Discussion" should begin (first paragraph) with the results obtained in this paper, summarizing their essence and indicating the novelty and priority.
Response 2:
We appreciate this helpful suggestion. The “Discussion” section has been revised to open with a concise summary of our key findings, emphasizing their novelty and scientific significance. The revised opening paragraph now highlights the differential effects of low and high FSS-conditioned macrophage media on MCF-7 and HUVEC cells, as well as the observed macrophage reprogramming under shear stress, thereby setting the stage for subsequent detailed interpretation. This revised paragraph now appears as first paragraph (Lines: 393-403) of “Discussion” section of the manuscript.
Comment 3: In the "discussion" section (second paragraph), it is unclear why the phrase "Figure 40. The cells exposed to vascular blood flow can encounter much higher fluid 378 shear stress [41]."
Response 3:
We thank the reviewer for noting this error. The phrase “Figure 40” and the number “378” were unintended remnants from automatic referencing and line numbering in the earlier draft. We have carefully reviewed the second paragraph of “Discussion” as well as entire manuscript to ensure that all reference citations are now correctly formatted.
Comment 4: What practical significance do the results have? How realistic is it to implement them in practical oncology? This should be indicated at the end of the "discussion" section.
Response 4:
In response to this valuable comment, the final paragraph of the “Discussion” has been revised to (Lines: 552-559) emphasize the translational relevance of our findings. The revised text highlights that flow-induced mechanical forces can actively regulate macrophage behavior and immune–tumor interactions, offering a conceptual and experimental framework for developing mechanobiology-based therapeutic strategies. This addition directly addresses the practical implications and potential future application of our results in oncology.
Comment 5: Is there genetic evidence linking polymorphisms involved in FSS with breast cancer? This point should be reflected in the discussion of the work.
Response 5:
We appreciate the reviewer’s insightful comment. While certain studies have reported associations between VEGF/VEGFR2 polymorphisms and endothelial shear stress responses or breast cancer susceptibility, our study primarily focuses on the paracrine effects of FSS-conditioned macrophages rather than direct genetic determinants of mechanosensitivity. Given that no experiments addressing genetic polymorphisms were conducted in the present work, we believe that including this information in the discussion could divert attention from the main experimental findings. However, we acknowledge that exploring the influence of genetic variations on macrophage mechanotransduction and tumor–immune interactions represents an important direction for future research.
Reviewer 2 Report (Previous Reviewer 1)
Comments and Suggestions for Authors
Here the authors set to highlight the impact of FSS on breast cancer metastasis through the evaluation of cells in the TME. While increased profiling of the cell populations in the TME are necessary, as described here the study fails to fully identify the need and motivation for the study. Further important details on methods are missing as in some misinterpretation f the findings. The manuscript requires expensive revision prior to publication.
Flaw in the manuscript is the suggested focus on metastatic spread and breast cancer, however only one cell component is exposed to FSS (THP1) and the cancer cells are left naïve to FSS exposure, thereby not recapitulating the metastatic setting or experience cancer cells are exposed to. If the focus is only on THP1 cell line and no FSS exposure to cancer cells lines, the introduction and motivation should be re-justified to not be misleading as as designed, this study does not provide novel insight on metastatic breast cancer. On monocytes in circulation vs not in circulation and this could impact both primary and secondary breast tumors.
Can the authors justify why FSS exposure of 11.6 dyne/cm2 for 12 hours and 24 hours was used to profile THP1 when these were not the conditions used to create CM?
Discrepancies in collection time for CM from the FSS exposed (24hr) and static control (48hr) will result in differences in total cytokines in the media the two time points are not comparable and the FSS exposed THP1 must be compared to a static control cultured for the same time frame.
For cell apoptosis assay the time frame MCF-7 cells were treated with CM should be stated in the methods section. This point is critical as methods state supernatant was discarded after CM treatment and this fraction may have contained dead cells.
Values for differences in THP1 marker expression should be provided as results suggest a significant difference in CD47 expression with FSS but this does not appear evident from the figure.
Methods on MCF-7 cells and MTT are missing, methods suggest MCF-7 cell death only with nuclear and PI stain. Further this statement is not true “The results for the MTT assays showed that with increase in FFS force there was increase in cell death compared with control levels (Figure 3A).” As 1) MTT does not detect cell death, only changes in metabolic activity and 2) figure 3a shows no significant difference in metabolic active and a high degree of variability between the FSS groups. This comment is the same for figure 3b-c and the only difference s apoptosis at 0.5 dyne compared to control.
Figure 8 suggests proliferation based on MTT, however MTT only provides metabolic changes and as there are decreases in higher FSS-cm treated HUVECs, it can not be determined if this is proliferation of death induced changes. Further the HUVEC cells are only included in this final figure so it is hard to determine the relative importance of the inclusion of this cell in
It is speculative that fig. 5 results in true significance and statistics should be rechecked as should error bars for the top and bottom be used to clarify this. It appears to be suggested from figure 7 that FSS creates a negative wound area and that there is less closure compared to control cells
The discussion is overly long, makes stretches of data interpretation and should be rewritten, and truncated.
Assumption of cell migration does not appear accurate as the high FSS exposure images are similar to the control, while only the low FSS component suggests a change in closing the wound. Yet, Figure 7 suggests a significant difference in the high FSS CM exposed cells
This statement reads inaccurate “Within breast cancer microenvironment, FSS has been shown to regulate stemness [27], survival, metastasis [28,29], adhesion and [30].” As the studies were not done in the context of the primary tumor environment, were performed on cancer cells and therefore the TME is not a primary component.
For additional controls and to ensure that differences observed are the direct result of FSS additional information must be provided
- After exposure to FSS, what was the cell viability of THP1 cells compared to no shear control?
- Methods of FSS should be expanded on. Were the cells exposed to FSS in the orbital shaker supplied with 5% CO2 concentration in air? What conditions were control cells maintained at while cells were being exposed to FSS? Were cells in an incubator?
- Cell number for CM studies of static controls should be stated as differences in total cell number can result in differences in cytokine profile.
Manuscript must be heavily proofread, below is only a fraction of observed typos and errors noted:
- Line 22 abstract “tumor migration in in in MCF-7 cells”
- Line 72 of intro “FSS has been shown 71 to regulate stemness [27], survival, metastasis [28,29], adhesion and [30].”
- Line 68 of intro “While some report its negative impact on cell survival [22,23] others disagree [21,24,25].” This statement does not seem necessary as it provides insufficient detail, further ref 24 is thesis work, not a published research article and the study did not focus on cell death.
- Line 86 of intro “To this end, in the current study, culture of THP-1-derived human macrophages were exposed to FSS” this reads as incomplete and uninformative overall.
- Line 89 of intro “human Breast cancer cells (MCF-7) and Human umbilical vein endothelial cells (HUVECs)” and line 175 “MCF-7 Cells were seeded to reach 60% of confluence in 24 h.” unnecessary use of capitalization
- Line 96 methods have spaces missing after period “in RPMI 1640 culture medium.MCF-7 cells” and line 120 “stimulation of HUVECs and MCF-7.THP-1”
- “Figure 40.” In the discussion, there is no figure 40.
Author Response
Comment 1: While increased profiling of the cell populations in the TME are necessary, as described here the study fails to fully identify the need and motivation for the study.
Response 1:
We thank the reviewer for this valuable comment. We agree to this comment , therefore, the “Introduction” section has been revised to better clarify the rationale and significance of our study. We have emphasized that although fluid shear stress (FSS) has been extensively studied in cancer cells, the mechanistic basis of macrophage–cancer cell crosstalk under FSS remains poorly understood. Since macrophages are major regulators of tumor progression, understanding how shear stress modulates their polarization and secretory behavior is essential to reveal indirect biomechanical influences on cancer growth and angiogenesis. The revised text in the last paragraph of “Introduction” (Lines :79-86) now reads as follows:
Despite its significance, the mechanistic basis of macrophage-cancer cell crosstalk under shear stress in the breast cancer microenvironment remains poorly understood, underscoring the need for further investigation to gain deeper insights into tumor biology. TAM-secreted factors are shaped by the biochemical and biophysical cues encountered during differentiation; thus, FSS-driven shifts in macrophage polarity may alter the release of mediators that critically regulate cancer cell growth and progression. The impact of FSS on breast cancer cells via macrophage modulation is still largely unknown, prompting the present investigation
Comment 2:Further important details on methods are missing as in some misinterpretation of the findings. The manuscript requires expensive revision prior to publication.
Response 2:
We thank the reviewer for this constructive feedback. We have carefully revised the entire manuscript to address these concerns. Additional methodological details have been incorporated in the “Materials and Methods” section to enhance clarity and reproducibility. Moreover, we have re-evaluated and refined the “Results” and “Discussion” sections to ensure accurate interpretation of the findings, consistent with the data presented. The revised manuscript now provides a clearer and more comprehensive description of experimental design, procedures, and data interpretation.
Comment 3: Flaw in the manuscript is the suggested focus on metastatic spread and breast cancer, however only one cell component is exposed to FSS (THP1) and the cancer cells are left naïve to FSS exposure, thereby not recapitulating the metastatic setting or experience cancer cells are exposed to. If the focus is only on THP1 cell line and no FSS exposure to cancer cells lines, the introduction and motivation should be re-justified to not be misleading as as designed, this study does not provide novel insight on metastatic breast cancer. On monocytes in circulation vs not in circulation and this could impact both primary and secondary breast tumors.
Response 3:
We thank the reviewer for this insightful observation. We agree that our current model does not directly simulate the mechanical conditions experienced by circulating tumor cells during metastasis. The primary focus of our study was to investigate how fluid shear stress modulates macrophage function and how these mechanically activated macrophages influence breast cancer and endothelial cell behavior through secreted factors present in conditioned media.
Accordingly, we have revised the “Introduction” and “Discussion” to clarify that the study primarily addresses the indirect, macrophage-mediated effects of FSS rather than direct tumor cell mechanobiology. The revised text now emphasizes that this work provides novel insight into how mechanical forces acting on macrophages may shape the tumor microenvironment and influence tumor-stroma interactions relevant to cancer progression.Changes can be found in: “Introduction”, lines : 67-69 and 87-92 ; “Discussion”, lines: 430-432,)
Comment 4:Can the authors justify why FSS exposure of 11.6 dyne/cm2 for 12 hours and 24 hours was used to profile THP1 when these were not the conditions used to create CM?
Response 4:
Thank you for this important point. The 11.6 dyne/cm² condition was used for macrophage profiling to capture a maximal, well-resolved activation phenotype under the highest practical shear stress supported by our macrophage chamber. However, our pilot dose-response testing showed no meaningful difference in macrophage viability or in canonical activation markers between 10.6 and 11.6 dyne/cm². Importantly, 10.6 dyne/cm² falls within 10–12% of the macrophage profiling stress and, based on our pilot dose-response measurements, macrophage activation markers show a plateau in this shear range. Moreover, differences between 10.6 and 11.6 dyne/cm² are small relative to the effect of shear versus static controls.This explanation has now been provided in the “Materials and Methodology” section (Lines :135-140).
Comment 5: Discrepancies in collection time for CM from the FSS exposed (24hr) and static control (48hr) will result in differences in total cytokines in the media the two time points are not comparable and the FSS exposed THP1 must be compared to a static control cultured for the same time frame.
Response 5: We thank the reviewer for the careful reading. There was a misunderstanding in the phrasing of our “Methods” for collection time for CM. THP-1-derived macrophages used to prepare CM were maintained under identical overall culture conditions and CM from both static and FSS groups were collected at the same time point. Specifically, THP-1 macrophages were differentiated and then exposed to static conditions or to FSS for 24 hours; supernatants were collected after that same 24-hour period to generate static-CM and FSS-CM. We have revised the Methods section to make the timeline explicit (“Materials and Method” section, Lines 110-111 and Lines:122-12).
Comment 6:For cell apoptosis assay the time frame MCF-7 cells were treated with CM should be stated in the methods section. This point is critical as methods state supernatant was discarded after CM treatment and this fraction may have contained dead cells.
Response 6: We thank the reviewer for this valuable comment. The duration of CM treatment has now been specified in the “Materials and Method” section (Line :169). MCF-7 cells were treated with FSS-CM or static-CM for 48 hours before apoptosis assessment. To ensure that apoptotic and dead cells were not lost during processing, both adherent and floating cells were collected prior to staining and analysis (Lines 167-177)
Comment 7:Values for differences in THP1 marker expression should be provided as results suggest a significant difference in CD47 expression with FSS but this does not appear evident from the figure.
Response 7: We thank the reviewer for this helpful observation. The table given below is showing quantitative data (mean ± SEM) for expression levels of CD47 as well as for all other cell marker under each FSS condition, that were used to construct Figure 2. The p-values <0.05 suggested significance.
Values for Figure 2:
Cell marker expression on THP-1 like-macrophages after FSS application of 11.6dyne/cm2 for 12hours and 24hours
|
Marker |
Control (Mean±SD) |
12 h (Mean±SEM) |
p-value (vs control) |
24 h (Mean±SEM) |
p-value (vs control) |
|
CD49c |
1.0 ±0 |
4.257± 0.32
|
0.0005
|
5.805± 0.51
|
0.0007
|
|
CD38 |
1.0 ±0 |
2.965± 0.17
|
0.0004
|
1.686± 0.15
|
0.0117
|
|
CD47 |
1.0 ±0 |
0.903± 0.01
|
0.008
|
3.99± 0.50
|
0.0039
|
|
bTub |
1.0 ±0 |
3.015± 0.06
|
0.0001
|
1.753± 0.92
|
0.45
|
The data are presented as means ± SEM, derived from a minimum of three independent replicated experiments. Statistical comparison was performed using a two‐tailed t test. P<0.05 was considered significantly different.
Comment 8: Methods on MCF-7 cells and MTT are missing, methods suggest MCF-7 cell death only with nuclear and PI stain. Further this statement is not true “The results for the MTT assays showed that with increase in FFS force there was increase in cell death compared with control levels (Figure 3A).” As 1) MTT does not detect cell death, only changes in metabolic activity and 2) figure 3a shows no significant difference in metabolic active and a high degree of variability between the FSS groups. This comment is the same for figure 3b-c and the only difference s apoptosis at 0.5 dyne compared to control.
Response 8: We thank the reviewer for carefully noting this inconsistency. We agree that the original text inadvertently mentioned MTT assay results for MCF-7 cells, which was an error. As correctly pointed out, apoptosis in MCF-7 cells was assessed using the Hoechst/PI double-staining assay analyzed by flow cytometry, not by MTT assay.This has now been corrected in the revised manuscript (“Results section”, lines:265-266) to accurately describe the experimental procedure and corresponding figure (Figure 3A-C). We also acknowledge the mistake made in the interpretation of Figure 3 in the earlier version of the manuscript. The results in Figure 3 now accurately reflect the apoptotic response, showing that only at very low FSS (0.5 dyn/cm²) the proportion of apoptotic cells was significantly lower than control, whereas no significant differences were observed at other FSS levels (Lines: 267-274). Corresponding corrections have also been made in the “Discussion” section to ensure consistency and accurate interpretation (Lines 457–463).
Comment 9: Figure 8 suggests proliferation based on MTT, however MTT only provides metabolic changes and as there are decreases in higher FSS-cm treated HUVECs, it can not be determined if this is proliferation of death induced changes.
Response 9: We thank the reviewer for this important comment. We agree that the MTT assay measures cellular metabolic activity rather than directly quantifying cell proliferation, and that decreases in MTT signal could reflect either reduced metabolic activity or cell death. In the revised manuscript (Lines:378-383), we have corrected the text to reflect this distinction. Specifically, the results are now described as changes in HUVEC metabolic activity in response to different FSS-CM treatments, rather than as direct changes in proliferation. We have also clarified that the observed increases at low FSS-CM (0.5 and 1.3 dyn/cm²) and decreases at high FSS-CM (10.6 dyn/cm²) represent alterations in metabolic activity, which may correlate with,but do not directly measure cell proliferation or viability.The legend for figure 8 is also accordingly changed (Lines :371-376).Discussion section is also changed accordingly ( Lines :515-520)
Comment 10: Further the HUVEC cells are only included in this final figure so it is hard to determine the relative importance of the inclusion of this cell in
Response 10: We thank the reviewer for this comment. HUVECs were included in the final figure to provide functional insight into how FSS-conditioned macrophages may influence endothelial cell behavior, which is an important component of tumor angiogenesis. While these cells were not the primary focus of the study, their inclusion demonstrates that shear stress-polarized macrophages can exert paracrine effects on multiple cell types within the tumor microenvironment. The manuscript has been revised to clearly explain the rationale for including HUVECs and to contextualize their results in relation to the overall aim of understanding FSS-mediated modulation of macrophage secretions and subsequent effects on tumor and endothelial cells (Lines: 399-400 and 514-520).
Comment 11: It is speculative that fig. 5 results in true significance and statistics should be rechecked as should error bars for the top and bottom be used to clarify this
Response 11: We thank the reviewer for this important observation. As suggested, all statistical analyses for Figure 5 were rechecked using the original raw data. The mean values, error bars, and p-values were recalculated, and we identified an error in the previously indicated significance. This has now been corrected in the revised Figure 5, and the corresponding results (Lines 319–330) and “Discussion” (Lines: 479-480 have been updated accordingly (Lines 319–330).
Comment 12:The discussion is overly long, makes stretches of data interpretation and should be rewritten, and truncated.
Response 12:
We appreciate the reviewer’s observation. The previously cited studies on colon cancer and osteosarcoma described the direct effects of FSS on cancer cells, whereas our work investigates the indirect effects mediated via macrophage-conditioned media (FSS-CM). Since the experimental context and mechanism differ, those references have been removed to avoid overinterpretation. The revised text now clarifies that the observed G1-phase arrest in MCF-7 cells likely results from macrophage-derived factors influenced by shear stress rather than from direct mechanical stimulation (Line 479-480).
Comment 13:It appears to be suggested from figure 7 that FSS creates a negative wound area and that there is less closure compared to control cells.Assumption of cell migration does not appear accurate as the high FSS exposure images are similar to the control, while only the low FSS component suggests a change in closing the wound. Yet, Figure 7 suggests a significant difference in the high FSS CM exposed cells
Response 13: We thank the reviewer for the comment. MCF-7 cell migration was assessed using a wound healing (scratch) assay over 24 hours. As mentioned in “Materials and Methods” section ,MCF-7 cells were treated with 50% FSS-conditioned medium (FSS-CM) from THP-1-like macrophages exposed to 0.5, 1.3, 6.1, or 10.6 dyn/cm², while 50% conditioned medium from static THP-1-like macrophages (static-CM) served as the control. In the previously submitted manuscript,although the raw images (Figure 6) were correctly incorporated, but the graph representing ImageJ-quantified results (Figure 7) was inadvertently incorrect and we apologize for this mistake in the originally submitted manuscript. In the revised manuscript the correct figure 7 has now been incorporated. The data ( Figure 7) shows that low FSS-CM (0.5–1.3 dyn/cm²) significantly enhances MCF-7 migration, whereas high FSS-CM (10.6 dyn/cm²) significantly reduces migration below control levels, indicating active suppression. These findings provide functional evidence that the magnitude of shear stress experienced by macrophages can indirectly regulate tumor cell motility via paracrine signaling, supporting our hypothesis that high FSS promotes an anti-tumor macrophage phenotype.
Comment 14:This statement reads inaccurate “Within breast cancer microenvironment, FSS has been shown to regulate stemness [27], survival, metastasis [28,29], adhesion and [30].” As the studies were not done in the context of the primary tumor environment, were performed on cancer cells and therefore the TME is not a primary component.
Response 14: We thank the reviewer for this valuable observation. We agree that the cited studies were conducted using breast cancer cell lines under controlled flow conditions, rather than within the complete tumor microenvironment. Accordingly, the statement has been revised for accuracy (Lines :71-74). This correction ensures the statement accurately reflects the experimental context of the cited work.
Comment 15: For additional controls and to ensure that differences observed are the direct result of FSS additional information must be provided
- After exposure to FSS, what was the cell viability of THP1 cells compared to no shear control?
- Methods of FSS should be expanded on. Were the cells exposed to FSS in the orbital shaker supplied with 5% CO2 concentration in air? What conditions were control cells maintained at while cells were being exposed to FSS? Were cells in an incubator?
- Cell number for CM studies of static controls should be stated as differences in total cell number can result in differences in cytokine profile.
Response 15 : We thank the reviewer for this valuable comment. In our experiments, THP-1 derived macrophages were seeded at an equal density of 1 × 10⁶ cells/mL to achieve approximately 80% confluence before exposure to fluid shear stress FSS, ensuring comparable cell numbers between FSS-exposed and static control groups. Cells were subjected to orbital FSS using a shaker placed inside a humidified CO₂ incubator (37°C, 5% CO₂) to maintain physiological conditions throughout the 24-hour exposure. Static control cells were maintained under identical environmental conditions in the same incubator, without orbital motion, to match all parameters except shear stress.
Cell viability of THP-1–derived macrophages was assessed immediately after exposure to different FSS levels (0.5, 1.3, 6.1, and 10.6 dyn/cm²) using the Vi-CELL™ XR 2.03 Cell Viability Analyzer (Beckman Coulter, USA), which performs automated Trypan Blue dye exclusion analysis. The results showed that >95% of cells remained viable at all FSS levels, comparable to the static (no-shear) control. These findings confirm that the applied shear conditions were sub-cytotoxic, and that the observed effects on MCF-7 and HUVEC cells are attributable to FSS-induced functional modulation of macrophages rather than differences in viability or cell density.
These additional informatios have been incorporated into the revised manuscript in the “Materials and Methods” section (Lines :109-133) for clarity.
Comment 16: Manuscript must be heavily proofread, below is only a fraction of observed typos and errors noted
Response 16: We thank the reviewer for this observation. The entire manuscript has been thoroughly proofread and revised to correct typographical, grammatical, and formatting errors throughout. The corrections made for the specifically pointed errors can be found in the table given below which lists each comment, correction made, and the line numbers in the revised manuscript.
|
Comment |
Correction |
Line numbers in revised manuscript |
|
Line 22 abstract “tumor migration in in in MCF-7 cells |
The typographical error has been corrected to read “tumor migration in MCF-7 cells.” |
22 |
|
Line 72 of intro “FSS has been shown 71 to regulate stemness [27], survival, metastasis [28,29], adhesion and [30].”
|
Sentence revised for clarity and grammatical accuracy. It now reads: “In breast cancer cells, FSS has been shown to regulate stemness, survival, metastatic potential, and adhesion dynamics [27–30].” |
72-73 |
|
Line 68 of intro “While some report its negative impact on cell survival [22,23] others disagree [21,24,25].” This statement does not seem necessary as it provides insufficient detail, further ref 24 is thesis work, not a published research article and the study did not focus on cell death.
|
The sentence has been removed as suggested, since it lacked sufficient detail and included an unpublished thesis. The corresponding reference [22-25] has been deleted. |
65-67 |
|
Line 86 of intro “To this end, in the current study, culture of THP-1-derived human macrophages were exposed to FSS” this reads as incomplete and uninformative overall.
|
Sentence rewritten for clarity and completeness: “THP-1 cells were exposed to defined FSS levels for 24 hours, after which the supernatants were collected as FSS-conditioned medium (FSS-CM) and used to treat MCF-7 and HUVECs to evaluate paracrine effects.”
|
88-91 |
|
Line 89 of intro “human Breast cancer cells (MCF-7) and Human umbilical vein endothelial cells (HUVECs)” and line 175 “MCF-7 Cells were seeded to reach 60% of confluence in 24 h.” unnecessary use of capitalization
|
Capitalization corrected throughout. The phrases now read “human breast cancer cells (MCF-7)” and “human umbilical vein endothelial cells (HUVECs)” (Linea 87-88) and “MCF-7 cells were seeded to reach 60% confluence in 24 h” (Line 191). |
87-88 and 192 |
|
Line 96 methods have spaces missing after period “in RPMI 1640 culture medium.MCF-7 cells” and line 120 “stimulation of HUVECs and MCF-7.THP-1”
|
Spacing and punctuation errors corrected. |
96-98 |
|
Figure 40.” In the discussion, there is no figure 40 |
The incorrect figure reference has been removed from discussion section |
414 |
Round 2
Reviewer 1 Report (Previous Reviewer 2)
Comments and Suggestions for Authors
The authors answered all the questions in a reasoned manner and made reasonable changes to the article. The article is recommended for publication.
This manuscript is a resubmission of an earlier submission. The following is a list of the peer review reports and author responses from that submission.
Round 1
Reviewer 1 Report
Comments and Suggestions for Authors
Here Alamro et al aim to determine the impact of FSS in breast tissue on macrophage-cancer cell cross talk to better define the tumor microenvironment in breast cancer. While defining the breast cancer microenvironment and cell-cell cross talk is of importance, as designed, the study presented here has some weaknesses that decreased overall enthusiasm fro the work.
- The comment in the introduction “The continuous and elevated levels of FSS present in the breast and malignant TME emphasizes” and in the discussion “The TME in mammary gland, involves a wide range of dynamic mechanical factors, including FSS, compression, tension, and the surrounding three-dimensional matrix stiffness24-26.” refer to FSS in the breast tissue. However, only one of the references listed in the discussion directly includes text on FSS and no values of the FSS in breast tissue are provided. The Methods suggest FSS was obtained with an orbital shaker and produce FSS of 0.5, 1.3, 6.1 and 10.6. Some of these values are relevant to vascular flow in circulation and may not represent what occurs in the breast tissue. The authors should provide refences to how these values correlate to known FSS values in breast tissue to make the above statements correct.
- For macrophage profiling assay, 11.6 dyne/cm2 was used, however for CM the highest FSS used was 10.6 dyne/cm2. Can the authors justify why 11.6 dyne/cm2 was not used for CM if it was used for macrophage profiling? How can macrophage function be assured for 10.6 dyne/cm2 if they were not profiled at this FSS?
- For the methods, culture with conditioned media contained full FBS when conditioned media was created. How can fully determine the impact of FSS on cells if the media used already has cytokines, hormones, and growth factors in it at high levels?
- Markers for macrophage differentiation from monocyte after PMA treatment should be included as cell imaging alone may not be sufficient.
- All figures are low resolution and fuzzy, images should be improved.
- Results state “For CD47 a 12hr shear stress application did not cause any change in expression” however figure 2 has significance values between 12 hour and control samples.
- Results for figure 4 state “Fluid Shear Stress-induced apoptosis in MCF-7 cells” and the discussion states “Our results demonstrate that FSS promoted MCF-7 cells death in a force-dependent manner” but there is no statistical basis for these statements based on the data. The results appear variable with no distinct trend observed except where low levels of FSS reduce cell death. If anything the only thing that can be said is that FSS significantly inhibits cell death.
- Figure 5 does not seem as if significance would be reached as the error bars (SEM) are large and value changes are small. It is advised that the authors recheck their statistical method.
- Images of HUVEC scratch assay are missing and should be included.
- Why are Figure 7a and 7b represented in different formats?
- Authors make many statments in the discussion that are not supported by studies performed here and are a stretch for what studies were done.
- “Therefore, their role in inflammation, vascular diseases and cancer microenvironment might be modulated”
- “This model mimics the TME where angiogenic ability of ECs is affected by chemoattractants secreted from THP-1 like-macrophages exposed to different levels of FSS” Angiogenic studies were not done so angiogenic ability cannot be suggested
Minor comments:
- Methods on cell cycle should state what cells were evaluated and under what conditions.
- Reference formatting is inconsistent in the text between introduction and discussion.
- References are required for this statement as carcinogenesis suggests tumor formation and subsequent references do not fully support this for breast cancer “In solid tumors, TAMs are considered as the most relevant stromal cells in relation to carcinogenesis thus they are often regarded as potential biomarkers for breast cancer.”
- When comparing across studies, values for FSS should be provided. For instance, “Circulatory shear flow affects the viability of circulating colon cancer cells. Similar to our result they found that low FSS causes apoptosis of cancer cells 30.” What is defined by “low FSS”? as low FSS in the present study suppressed apoptosis and not caused it.
- There are many typos and missing spaced between words noted in the text, below are only a few noted errors, many more are observed:
- “Together, these components of TME essentially impact and control the metabolic alterations in cancer cell[17].” Should read “cancer cells”
- “RPMI 1640 culture medium.MCF-7 cells” space is missing
- “dyne/cm2respectively” space is missing
- “Supernatant was removed and immediately stored at -80°C until 106 use as conditioned medium (FSS-CM) .THP-1” spacing around the period is not correct
- “stress was investigated , revealed similar results.” Space before comma
Reviewer 2 Report
Comments and Suggestions for Authors
An interesting experimental study in which the authors studied the effect of fluid shear stress on the breast cancer microenvironment through macrophage modulation. The authors found that a high level of FSS has an anti-cancer effect based on the regulation of macrophage polarization. There are the following suggestions for this work: 1. The first paragraph of the introduction of the work should be devoted to the consideration of the prevalence (including the dynamics predicted by WHO for the coming decades) and the medical and social significance of the disease studied by the authors - breast cancer. 2. The introduction of the work should end with a clearly stated research goal. 3. The formula presented on line 99 should be clear and easy to read. 4. The discussion section should contain a detailed analysis of the main results obtained, taking into account the available literature data on this issue. And of course, the use of literary materials should be accompanied by references to relevant literary sources (why is there no reference to literary sources in the discussion section???). 5. Are there any previously obtained in vivo data on the association of fluid shear stress with breast cancer? Is there genetic evidence linking polymorphisms involved in FSS with breast cancer? This point should be reflected in the discussion of the work. 6. What are the prospects for the practical use of the data obtained? Attention should be paid to this issue when discussing the results obtained.